# An Inter-Laboratory Comparative Study on the Influence of Reagents to Perform the Identification of the *Xylella fastidiosa* Subspecies Using Tetraplex Real Time PCR

Nicoletta Pucci [1,*], Valeria Scala [1], Erica Cesari [1], Valeria Crosara [1,2], Riccardo Fiorani [1], Alessia L'Aurora [1], Simone Lucchesi [1], Giuseppe Tatulli [1], Eleonora Barra [3], Serena Ciarroni [4], Francesca De Amicis [5], Salvatore Fascella [6], Francesca Giacobbi [7], Francesca Gaffuri [8], Valeria Gualandri [9], Lucia Landi [10], Giuliana Loconsole [11], Giulia Molinatto [12], Stefania Pollastro [13], Maria Luisa Raimondo [14], Domenico Rizzo [15], Chiara Roggia [16], Maria Rosaria Silletti [17], Simona Talevi [18], Marco Testa [19] and Stefania Loreti [1]

1   Council for Agricultural Research and Economics, Research Centre for Plant Protection and Certification (CREA-DC), 00156 Rome, Italy; valeria.scala@crea.gov.it (V.S.); erica.cesari@crea.gov.it (E.C.); valeria.crosara@uniroma1.it (V.C.); riccardo.fiorani@crea.gov.it (R.F.); alessia.laurora@crea.gov.it (A.L.); simone.lucchesi@crea.gov.it (S.L.); giuseppe.tatulli@crea.gov.it (G.T.); stefania.loreti@crea.gov.it (S.L.)
2   Department of Environmental Biology, Sapienza University of Rome, 00185 Rome, Italy
3   Phytopathological Laboratory, Campania Region, 80141 Napoli, Italy; eleonora.barra@unina.it
4   Phytoparasites Diagnostics (Phy.Dia.) s.r.l., 01100 Viterbo, Italy; info@phydia.eu
5   Regional Agency for Agricultural Development—Plant Protection Service of Friuli Venezia Giulia Region, 33050 Pozzuolo del Friuli, Italy; francesca.deamicis@ersa.fvg.it
6   Plant Health Protection Service, Liguria Region, 16129 Genova, Italy; salvatore.fascella@regione.liguria.it
7   Regional Plant Protection Service, 40128 Bologna, Italy; francesca.giacobbi@regione.emilia-romagna.it
8   Laboratory of Plant Health Service, 22070 Vertemate con Minoprio, Italy; francesca_gaffuri_cnt@regione.lombardia.it
9   Technology Transfer Centre, Fondazione Edmund Mach, 38010 San Michele all'Adige Trento, Italy; valeria.gualandri@fmach.it
10  Department of Agricultural, Food and Environmental Sciences, Marche Polytechnic University, 60131 Ancona, Italy; l.landi@univpm.it
11  National Research Council, Institute for Sustainable Plant Protection (IPSP)—National Research Council, 70126 Bari, Italy; giuliana.loconsole@ipsp.cnr.it
12  Phytosanitary Laboratory of the Phytosanitary Sector and Technical, Scientific Services of the Piedmont Region, 10144 Torino, Italy; giulia.molinatto@regione.piemonte.it
13  Department of Soil, Plant and Food Sciences (DISSPA), University of Bari, 70126 Bari, Italy; stefania.pollastro@uniba.it
14  Department of Agricultural Sciences, Food, Natural Resources and Engineering, University of Foggia, 71122 Foggia, Italy; marialuisa.raimondo@unifg.it
15  Laboratory of Phytopathological Diagnostics and Molecular Biology, Plant Protection Service of Tuscany, 51100 Pistoia, Italy; domenico.rizzo@regione.toscana.it
16  Laboratory Enocontrol Scarl Agri-Food Research Analysis Center, 12051 Alba, Italy; c.roggia@enocontrol.com
17  Research, Experimentation and Education Centre in Agriculture "Basile-Caramia", Locorotondo, 70010 Bari, Italy; mariarsilletti@crsfa.it
18  Plant Protection Service Marche Region—AMAP, 60027 Osimo, Italy; talevi_simona@assam.marche.it
19  Agris Sardegna, Agricultural Research Agency of Sardinia, 09020 Sassari, Italy; mtesta@agrisricerca.it
*   Correspondence: nicoletta.pucci@crea.gov.it

**Abstract:** In 2022, a test performance study (TPS) assessing the influence of different master mixes on the performance of the tetraplex real-time PCR (TqPCR) assay was organized. TqPCR allows for the specific detection and identification of *Xylella fastidiosa* (*Xf*) subspecies in a single reaction. Eighteen official laboratories of the Italian National Plant Protection Organization received a panel of 12 blind samples, controls, primers, probes, and different master mixes to participate in the TPS. Furthermore, the Research Centre for Plant Protection and Certification of the Council for Agricultural Research and Economics performed an intra-laboratory study (ITS) on spiked plant matrices to evaluate the analytical sensitivity of TqPCR employing the selected master mixes with the best performance. Naturally infected samples were analyzed for subspecies identification via TqPCR compared with the

official multilocus-sequence-typing (MLST) method. The best results in this comparative study were obtained using Fast Universal PCR Master Mix (Applied Biosystems) and Brilliant multiplex QPCR Master Mix (Agilent), and they confirmed that the TqPCR test is reliable, offering the advantage of identifying this subspecies at the same time, thus saving time and resources. The TqPCR assay is suggested among the tests to be used by laboratories performing the official diagnosis of *Xf* to support the activities of official monitoring.

**Keywords:** test performance study (TPS); intra-laboratory study (ITS); *Xylella fastidiosa* subspecies; tetraplex real-time PCR; critical reagent; performance criteria

## 1. Introduction

*Xylella fastidiosa* (*Xf*) is a Gram-negative bacterium belonging to the *Lysobacteraceae* (formerly *Xanthmonadaceae*) family and it inhabits the xylem vessels of about 679 plant species from 88 botanical families [1]. This causal agent of significant plant diseases (e.g., Pierce's disease in grapevine, citrus variegated chlorosis, olive quick decline syndrome (OQDS), and phony peach disease) is transmitted by xylem-sap-feeding hemipteran vector insects. In susceptible plant hosts, symptoms of *Xf* include leaf marginal necrosis and abscission, dieback, and death due to the obstruction of the xylem and a lack of sufficient water and nutrient flow because of bacterial growth (cell–cell adhesion and biofilm formation) within the plant vessels and the host defense response (tyloses and gum production) [2]. *Xf* has been included in the EPPO A2 list since 2017. In the European Union (EU), according to the Commission Implementing Regulation of (EU) 2019/2072 [3], *Xf* is included in the List of Union quarantine pests and their respective codes, Annex II, Part B: Pests Known to Occur in the Union Territory. Since its first outbreak in the Apulia region in Southern Italy [4], where *Xf* currently causes devastating OQDS in *Olea europaea*, inflicting severe socioeconomic and ecological damage, the pathogen continues to spread and has successfully established itself in some European countries (Corsica, Provence-Alpes-Côte d'Azur, and Occitaine in France; the Balearic Islands, Madrid, and Comunitat Valenciana in Spain; and in the Centro, Norte, and Lisboa e Vale do Tejo regions in Portugal [5,6]. In Italy, *Xf* was reported in the Tuscany region in 2018 and in the Lazio region in 2021 [1]. According to forecast models, the changing Mediterranean climate will lead to a potential expansion of the area of spread in most of Italy, Corsica, Spain, Portugal, Albania, Montenegro, Greece, and Turkey as well as all North African and Middle Eastern countries [7]. Based on genome sequence analyses, three subspecies have been described, namely, *fastidiosa*, *multiplex*, and *pauca*, which are associated with Pierce's disease of grapevine, citrus variegated chlorosis, and various leaf scorch diseases (including OQDS), respectively; the subspecies *sandyi* and *morus* were recently included within the subspecies *fastidiosa* [8,9].

As soon as a new outbreak is detected in a previously *Xf*-free area or in association with a new host plant, the identification of the corresponding subspecies and sequence type (ST) allows for the study of the recombination events that drive the genetic variability of the pathogen [8]. The information on the subspecies established in an area defines the list of plant species to be eradicated in order to prevent the spread of a pathogen. The host plant is closely related to the *Xf* subspecies, and only 15 host species can be infected by all *Xf* subspecies [1]. Therefore, the identification of this subspecies with a reliable, rapidly acting, and accurate tool directly applicable to plant samples is relevant for the application of phytosanitary measures by National Plant Protection Organization (NPPO).

The following molecular tests are recommended by EU legislation (Commission Implementing Regulation EU 2020/1201) for the assignment of subspecies [10] and by EPPO PM 7/24 (5) [11] for subspecies identification in the case of a new finding (i.e., a new outbreak or new hosts): (i) conventional PCR method developed by the Pooler and Hartung [12] and Hernandez Martinez [13] for the determination of the subspecies *pauca* and *multiplex-sandyi*, respectively, and (ii) Sanger sequencing of the PCR products of two house-

keeping genes (*malF*/*cysG* or *malF*/*rpodD*) sequenced in both directions. Moreover, strains can be classified into STs based on the multi-locus-sequence-typing analysis (MLST) [14] of seven individually amplified housekeeping genes (2-isopropylmalate synthase -*leuA*; ubiquinol cytochrome c oxidoreductase C1 subunit -*petC*; ABC transporter sugar permease -*malF*; sirohaem synthase -*cysG*; DNA polymerase III holoenzyme chi subunit -*holC*; NADH ubiquinone oxidoreductase NQO12 subunit -*nuoL*; and glutamate symport protein -*gltT*). To date, 90 sequence types have been described [15]. In the case of inconsistent results for the two sequenced genes or atypical/new patterns, a complete MLST analysis of the seven genes should be performed, and the sequences should be compared with data available in the pubMLST database (as suggested by EPPO PM 7/24 (5) [11]). MLST characterization can be performed for *Xf* via the direct amplification of DNA extracted from plant samples due to the difficulties in the pathogen isolation procedure. However, in the case of a low amount of DNA or the presence of inhibitor compounds, MLST analysis may fail.

The tetraplex real-time PCR (TqPCR) method developed by Dupas et al. [16] allows for *Xf* subspecies identification directly from plant samples, providing a robust and modular test for mixed infections as well [17]. These assays allow for the detection and identification of *Xf* subspecies with up to 10–100 pg/mL of DNA and $1 \times 10^3$–$1 \times 10^4$ colony-forming units (CFU) mL$^{-1}$ in spiked samples [16]. The sensitivity in spiked samples is better than the $10^5$ CFU mL$^{-1}$ sensitivity of MLST analysis [14,17] and is similar [16] to the sensitivity of the reference real-time PCR method developed by Harper et al. [18]. These multiplex real-time PCR assays offer a new, faster, more reliable, specific, sensitive, and less-expensive tool than MLST [14]. The assay includes six primer and probe combinations ("Set") specific for the different *Xf* subspecies. In particular, Set N° 2 allows for the detection of *Xf* and the identification of the subspecies *fastidiosa*, *multiplex*, and *pauca*, which are the subspecies widespread in Europe, but not of the subspecies *fastidiosa sensu lato*, *sandyi*, and *morus*.

In this study, the following approaches were used to evaluate the suitability of TqPCR for *Xf* diagnosis and subspecies determination:

(1) A test performance study (TPS), with the participation of 18 official laboratories (OLs) of the Italian (NPPO), was used to evaluate the influence of different master mixes on the performance of Set N° 2 of TqPCR [16]. TqPCR was compared with the real-time PCR method developed by Harper et al. [18], which is known as one of the most appropriate tests for the detection of *Xf*, with high diagnostic sensitivity and specificity [19].

(2) Intra-laboratory study (ITS), within the Research Centre for Plant Protection and Certification of the Council for Agricultural Research and Economics (CREA-DC), was used to evaluate the analytical sensitivity (ASE) of TqPCR [16] with respect to Harper et al. [18] and Hodgetts et al. [20] assays by testing spiked plant matrices. Additionally, analyses of naturally infected samples to determine the subspecies and STs via MLST [14] and TqPCR [16] were performed.

## 2. Materials and Methods

### 2.1. Test Performance Study (TPS)

A TPS was carried out in 2022 by incorporating 18 OLs of the Italian NPPO to comply with Article 101(c) of Regulation (UE) 2017/625. OLs are authorized by the NPPO to routinely perform official analysis for the national monitoring activity. Thirty-three sets of samples consisting of nucleic acid (DNA) were prepared, of which twenty-three sets were shipped to OL participants and 10 sets were employed by CREA-DC, the organizing laboratory, for the verification of homogeneity and stability. Each set consisted of 12 DNA samples; 3 positive amplification controls (PAC1, PAC2, and PAC3) of *Xf* suspensions of strains from the subspecies *fastidiosa*, *multiplex*, and *pauca*, respectively; and 1 negative amplification control (NAC) consisting of molecular-grade water. The samples consisted of: (i) olive (*Olea europaea*), vine (*Vitis vinifera*), lavender (*Lavandula* spp.), and rosemary (*Rosmarinus officinalis*) DNA artificially contaminated (spiked) or not with known DNA concentrations of *Xf* strains of the different subspecies and (ii) naturally infected olive

and almond (*Prunus dulcis*) DNA. Artificially contaminated samples were spiked with 0.6 pg/µL and 6 pg/µL of *Xf* DNA (Table 1). Details on the samples' compositions are provided in Table 1.

**Table 1.** List of samples (S1–S12) prepared for the test performance study, positive controls (PAC 1-3), negative control (NAC), and their phytosanitary status. For each sample, the pathogen concentration and the plant matrix used to prepare the samples are reported. *Xfm* = *Xylella fastidiosa* subspecies *multiplex*; *Xff* = *Xylella fastidiosa* subspecies *fastidiosa*; *Xfp* = *Xylella fastidiosa* subspecies *pauca*.

| Sample ID | Sample Type (DNA Extract) | Phyto-Sanitary Status | Host |
|---|---|---|---|
| S1 | Healthy | Negative | Petioles and vine leaves (*Vitis vinifera*) |
| S2 | Healthy | Negative | Petioles and olive leaves (*Olea europaea*) |
| S3 | Artificially contaminated (*Xfm*-6 pg/µL) | Positive | Petioles and lavender leaves (*Lavandula* spp.) |
| S4 | Artificially contaminated (*Xfm*-0.6 pg/µL) | Positive | Petioles and rosemary leaves (*Rosmarinus officinalis*) |
| S5 | Artificially contaminated (*Xff*-0.6 pg/µL) | Positive | Petioles and vine leaves |
| S6 | Artificially contaminated (*Xfp*-0.6 pg/µL) | Positive | Petioles and lavender leaves |
| S7 | Healthy | Negative | Petioles and lavender leaves |
| S8 | Healthy | Negative | Petioles and rosemary leaves |
| S9 | Artificially contaminated (*Xfm*-0.6 pg/µL) | Positive | Petioles and lavender leaves |
| S10 | Naturally infected (*Xfm*) | Positive | Petioles and almond leaves (*Prunus dulcis*) |
| S11 | Naturally infected (*Xfp*) | Positive | Petioles and olive leaves |
| S12 | Artificially contaminated (*Xff*-6 pg/µL) | Positive | Petioles and vine leaves |
| PAC1 | Bacterial DNA (*Xff*-60 pg/µL) | Positive | Bacterial strain |
| PAC2 | Bacterial DNA (*Xfm*-60 pg/µL | Positive | Bacterial strain |
| PAC3 | Bacterial DNA (*Xfp*-60 pg/µL) | Positive | Bacterial strain |
| NAC | Water DEPC (Diethyl pyrocarbonate) | Negative | - |

Samples were randomized within each set, and the sets were randomly assigned to each OL. After the randomization process, each sample was labeled with a code consisting of the Lab ID and the sample number. Eighteen OLs usually involved in the official analyses of *Xf* participated in the TPS (Table 2). Primers, probes, reagents for the execution of the TqPCR, and one set of samples were shipped to the OLs in dry ice.

**Table 2.** List of participating official laboratories in the test performance study. * The Research Centre for Plant Protection and Certification of the Council for Agricultural Research and Economics (CREA-DC) served as the organizing laboratory and contributed six different groups of operators, indicated as CREA-DC 1, CREA-DC 2, CREA-DC 3, CREA-DC 4, CREA-DC 5, and CREA-DC 6.

| |
|---|
| CREA-DC Centro difesa e certificazione—Roma * |
| Agenzia Settore Agroalimentare delle Marche—SFR Regione Marche, Osimo (AN) |
| Agenzia Agris, Ussana (SU) |
| CNR—Istituto per la Protezione Sostenibile delle Piante, Bari (BA) |
| CRSFA, Centro di Ricerca, Sperimentazione e Formazione in Agricoltura "Basile Caramia", Locorotondo (BA) |
| DAFNE, Dipartimento di Scienze Agrarie degli Alimenti, Risorse Naturali e Ingegneria—Università degli Studi di Foggia, Foggia (FG) |
| DAFNE, Università degli Studi della Tuscia, Dipartimento di Scienze agrarie e forestali, Viterbo (VT) |
| Dipartimento di Scienze Agrarie, Alimentari ed Ambientali, Università Politecnica delle Marche, Ancona (AN) |
| Enocontrol S.c.a.r.l. Centro d'analisi e ricerca, Cuneo (CN) |
| ERSA, Laboratorio di Fitopatologia e Biotecnologie, Udine (UD) |

**Table 2.** *Cont.*

| |
|---|
| Fondazione Edmund Mach, San Michele all'Adige (TN) |
| Laboratorio Fitopatologico Regione Emilia-Romagna, Bologna (BO) |
| Laboratorio Fitopatologico Regione Liguria, Genova (GE) |
| Laboratorio Fitopatologico Regione Campania, Napoli, (NA) |
| Laboratorio fitosanitario e settore fitosanitario e servizi tecnico-scientifici, Direzione Agricoltura e cibo, Regione Piemonte, Torino (TO) |
| Laboratorio SFR Regione Lombardia, Vertemate con Minoprio (MI) |
| Regione Toscana SFR e di vigilanza del controllo agroforestale—Laboratorio Fitopatologico Regionale, Pistoia (PT) |
| SELGE—Dipartimento di scienze del suolo, della pianta e degli alimenti—Università di Bari (BA) |

### 2.1.1. Plant Material

The healthy plant material for sample preparation was collected in *Xf*-free areas (Rome, Italy). The absence of the pathogen was evaluated using real-time PCR (Harper et al. [18]) conducted according to the EPPO PM 7/24 (5) [11]. The two naturally infected samples, almond and olive, were collected in the infected areas of Viterbo province and Lecce province (Italy), respectively. A positive phytosanitary status was ascertained using real-time PCR (Harper et al. [18]) conducted according to the EPPO PM 7/24 (5) [11].

### 2.1.2. Bacterial Strains

Bacterial cultures of *Xf* subspecies *multiplex* strain CFBP 8416, *Xf* subspecies *fastidiosa* strain Temecula 1 NCPPB 4605, and *Xf* subspecies *pauca* strain CFBP 8402 were grown in PD2 medium [11] at 28 °C for 7–15 days depending on the strain.

### 2.1.3. Sample Preparation and DNA Extraction

DNA was extracted using the Gentra Puregene Yeast/Bact. Kit (Qiagen, Venleo, The Netherlands) for bacterial suspensions and using the QuickPick™ SML Plant DNA kit (QRET Technologies Ltd., Turku, Finland), associated with the automated platform

KingFisher™ mL Purification System (Thermo Fisher, Vantaa, Finland), for healthy plant matrices as recommended in EPPO PM 7/24 (5) [11], without performing an ultrasonication step. Naturally infected samples were processed using the DNeasy Mericon Food Kit (Qiagen, Redwood City, CA, USA). DNA concentrations were evaluated using Qubit (dsDNA HS Assay kit, Invitrogen Carlsbad, CA, USA). The DNA extracted from healthy plants was appropriately mixed with the DNA of *Xf* subspecies (Table 1) at a known concentration to obtain artificially contaminated samples.

### 2.1.4. Sample Homogeneity and Stability Test

The samples were tested for their homogeneity and stability according to the EPPO standard 7/122(2) [21] before shipment. Ten randomly selected sets were tested in duplicate for homogeneity using the real-time PCR method developed by Harper et al. [18], which was conducted according to the EPPO standard 7/24 (5) [11], with the following modification: the use of the enzyme SsoAdvanced Universal Probes Supermix (Bio-Rad, Hercules, CA, USA) without BSA (bovine serum albumin). Three randomly chosen aliquots were tested after 1 week of storage at temperatures of $<-15\ °C$, 2–8 °C, and 25 °C for mid-term stability and at the deadline of TPS (after 4 weeks) at the temperature of $<-15\ °C$ for long-term stability. The samples were considered sufficiently homogeneous and stable when the values of the standard deviation (SD) of the quantification cycles (Cq) were SD < 1.

### 2.1.5. Tetraplex Real-Time PCR (TqPCR)

OLs were divided into five different groups (named A, B, C, D, and E) according to the different master mixes employed to perform TqPCR [16] (Table 3). In the EPPO PM 7(24) 5 [11], Bio-Rad master mix (SsoAdvanced™ Universal Probes Supermix) is reported to have an annealing temperature of 60 °C; the annealing temperature of 63 °C was selected on the basis of a satisfactory preliminary internal laboratory comparison. Applied Biosystems (Foster City, CA, USA) master mix (TaqMan™ Fast Universal PCR Master Mix) is one of the most-used enzymes for amplification purposes; Qiagen (QuantiNova Pathogen + IC kit) and Agilent (Santa Clara, CA, USA) (Brilliant Multiplex qPCR Master Mix) master mixes have been optimized for multiplex real-time PCR.

**Table 3.** List of enzyme/reaction conditions used for the tetraplex real-time PCR (Dupas et al. [16]) conducted by the five participating official laboratory (OL) groups: each group is coded by a letter (A to E). In the table, the numbers of OLs that participated in each group are indicated.

| Group Code and Total Number of OLs | Enzyme | Step of Amplification Protocol | Cycles | Time | Temperature °C |
|---|---|---|---|---|---|
| A 6 OLs | SsoAdvanced™ Universal Probes Supermix (Bio-Rad) | Denaturation | 1× | 3 min | 95 |
| | | Amplification/ Fluorescence detection | 40× | 15 s | 95 |
| | | | | 30 s | 60 |
| B 5 OLs | SsoAdvanced™ Universal Probes Supermix (Bio-Rad) | Denaturation | 1× | 3 min | 95 |
| | | Amplification/ Fluorescence detection | 40× | 15 s | 95 |
| | | | | 30 s | 63 |
| C 4 OLs | QuantiNova Pathogen + IC kit (Qiagen) | Denaturation | 1× | 2 min | 95 |
| | | Amplification/ Fluorescence detection | 40× | 15 s | 95 |
| | | | | 30 s | 60 |
| D 4 OLs | TaqMan™ Fast Universal PCR Master Mix (Applied Biosystems™) | Enzyme activation | 1× | 2 min | 50 |
| | | Denaturation | 1× | 10 min | 95 |
| | | Amplification/ Fluorescence detection | 40× | 15 s | 95 |
| | | | | 30 s | 60 |
| E 4 OLs | Brilliant Multiplex qPCR Master Mix (Agilent) | Denaturation | 1× | 10 min | 95 |
| | | Amplification/ Fluorescence detection | 40× | 15 s | 95 |
| | | | | 1 min | 60 |

Each master mix was tested by groups of four or five laboratories, as reported in Table 3. The amplification conditions were set according to the manufacturer's recommendations, except for the Bio-Rad master mix, for which two annealing temperatures were tested (Group A: 60 °C; Group B: 63 °C). The amplification reactions (Table S1) were carried out as indicated by Dupas et al. [16]. The Set 2 primers and probes of TqPCR provided to the participants is shown in Table 4.

**Table 4.** Primers and probes of Set 2 provided to the participating official laboratories for the execution of the tetraplex real-time PCR method developed by Dupas et al. [16]: 1-XF for detecting *Xylella fastidiosa*; 2-XFF for detecting *Xylella fastidiosa* subspecies *fastidiosa*; 3-XFM for detecting *Xylella fastidiosa* subspecies *multiplex*; 4-XFP for detecting *Xylella fastidiosa* subspecies *pauca*.

| Target Species | Primers and Probe | Sequence |
|---|---|---|
| *X. fastidiosa* | 1-XF-F<br>1-XF-R<br>1-XF-Probe | 5′-AAC CTG CGT GAC TCT GGT TT-3′<br>5′-CAT GTT TCG CTG CTT GGT CC-3′<br>5′-FAM-GCT CAG GCT GAC GGT TTC ACA GTG CA-BHQ1-3′ |
| *X. fastidiosa* subspecies *fastidiosa* | 2-XFF-F<br>2-XFF-R<br>2-XFF-Probe | 5′-TTA CAT CGT TTT CGC GCA CG-3′<br>5′-TCG GTT GAT CGC AAT ACC CA-3′<br>5′-HEX-CCC GAC TCG GCG CGG TTC CA-BHQ1-3′ |
| *X. fastidiosa* subspecies *multiplex* | 3-XFM-F<br>3-XFM-R<br>3-XFM-Probe | 5′-ACG ATG TTT GAG CCG TTT GC-3′<br>5′-TGT CAC CCA CTA CGA AAC GG-3′<br>5′-ROX- ACG CAG CCC ACC ACG ATT TAG CCG-BHQ2-3′ |

**Table 4.** *Cont.*

| Target Species | Primers and Probe | Sequence |
|---|---|---|
| *X. fastidiosa* subspecies *pauca* | 4-XFP-F<br>4-XFP-R<br>4-XFP-Probe | 5′-TGC GTT TTC CTA GGT GGC AT-3′<br>5′-GTT GGA ACC TTG AAT GCG CA-3′<br>5′-CY5-CCA AAG GGC GGC CAC CTC GC-BHQ2-3′ |

### 2.1.6. Performance Criteria Evaluation

The performance of each master mix was evaluated in terms of diagnostic sensitivity (DSE), diagnostic specificity (DSP), and accuracy (ACC), for which the percentages of true negative (TN), false positive (FP), false negative (FN), and true positive (TP) results provided by the OLs were calculated [21,22]. Reproducibility was calculated according to the method reported by Langton et al. [23], considering samples 1, 2, 4, 5, 6, 7, and 8 with a pathogen concentration at the limit of detection [21].

### 2.1.7. Outliers

Data were considered outliers, and excluded from analysis, if (a) results of controls were non-concordant with their phytosanitary status; (b) accuracy was statistically different from the average of accuracy obtained by all laboratories; or (c) results were incomplete (e.g., no technical replicates were reported).

### 2.2. Intra-Laboratory Study (ITS)

The bacterial suspensions of *Xf* subspecies *fastidiosa*, *multiplex*, and *pauca* were added to the plant matrices of *O. europaea*, *V. vinifera*, and *P. dulcis*, respectively, at final concentrations of $10^6$, $10^5$, $10^4$, $10^3$, $10^2$, and 10 CFU mL$^{-1}$ [24]. DNA of spiked samples was extracted as reported in Section 2.1.3. Three independent biological replicates were prepared for each bacterial concentration, and each of the three biological replicates was run in technical triplicates ($n = 9$) in real-time PCR. The tests [16,18,20] were performed according to the EPPO PM 7/24 (5) [11], and the ASE values were determined for each test. Subspecies and ST identification was performed in 2022–2023 on two samples of asymptomatic olive trees

collected in the infected area of the Apulia region and on six samples of *Spartium junceum* and one of *Prunus dulcis* collected in the Latium region in the infected area of the Viterbo province. Seven housekeeping genes (*cysG*, *gltT*, *holC*, *leuA*, *malF*, *nuoL*, and *petC*) were individually amplified [12] following the protocol described in EPPO PM 7/24 (5) [11], with some modifications. The reaction conditions were as follows: the PCR mix (50 uL) was composed of 10 µL of Colorless GoTaq Flexi Buffer (5*), 3 µL of MgCl$_2$ Solution (25 mM), 1.5 µL each of primer (10 µM), 1 µL of dNTPs (10 mM), 0.8 µL of BSA (non-acetylated) (50 µg µL$^{-1}$), 0.3 µL of GoTaq G2 Hot Start Polymerase (5 U µL$^{-1}$), 5 µL of DNA, and 26.9 µL of molecular-grade water; the amplification conditions were 95 °C for 3 min, 35 cycles of 95 °C for 30 s, 60 °C for 30 s and 72 °C for 60 s, and a final step of 72 °C for 10 min. PCR products were separated on 1% agarose gels at 80 V for 90 min. All amplicons of the expected sizes (*leuA* = 708 bp, *petC* = 533 bp, *malF* = 730 bp, *cysG* = 600 bp, *holC* = 379 bp, *nuoL* = 557 bp, and *gltT* = 654 bp) were purified using ISOLATE II PCR and Gel Kit (Bioline). Afterwards, the amplicons were sequenced using Sanger sequencing, and the consensus sequences were assembled for each sample using Geneious prime 2022.1. (http://www.Geneious.com/basic, accessed on 17 January 2023). The obtained sequences were aligned with available sequences from http://pubmlst.org/xfastidiosa/ (accessed on 17 January 2023) to establish the alleles for each of the seven genes and to define the allelic profiles of the different samples. The same samples were assessed via TqPCR [16].

**3. Results**

*3.1. Test Performance Study (TPS)*

3.1.1. Participants

All the OLs were able to submit their results by the deadline, and the corresponding details are reported in Figure S1. One OL of group B was considered an outlier and excluded from the data analysis due to the inability to determine the Cq value with fluorophore Cy5 (incomplete results). Three OLs, one for group B, one for group C, and one for group D, gave negative results for the PAC; in particular, non-concordant results (false positive) were provided for *Xf* subspecies *fastidiosa* (group B) and *Xf* subspecies *pauca* (groups C and D). Their results were excluded from the overall evaluation.

3.1.2. Sample Homogeneity and Stability Test

The results of the homogeneity and stability tests (mid-term and long-term) for all the tested samples are reported in Tables S2 and S3. The results were in accordance with the phytosanitary status (healthy or contaminated) of the samples; the value of ΔCq and SD were in the acceptable range of variation for all the samples. The obtained values were considered sufficiently homogenous and stable, indicating that shipment and storage did not negatively influence the phytosanitary status of the samples.

3.1.3. Evaluation of Performance Criteria

Table 5 shows the performance criteria values for the different master mixes used; in particular, DSE, DSP, and ACC values of 100% were obtained using Universal PCR (Applied Biosystems, group D) and Brilliant multiplex QPCR Master Mix (Agilent, group E) for all the subspecies tested. The lowest ACC (89.6%) was observed when the master mix QuantiNova pathogen+IC kit (QIAGEN S.r.l, group C) was used for *Xf* detection. Small variations from the expected results were obtained using the master mix SsoAdvanced™ Universal Probes Supermix (Bio-Rad, group A and B) a both 60 °C and at 63 °C annealing temperatures (Table 6). The DSE of the master mixes ranged from 87.5 to 100%, while the DSP ranged from 93.7 to 100%.

**Table 5.** Value of diagnostic sensitivity (DSE), diagnostic specificity (DSP), and accuracy (ACC) obtained with the different tetraplex real-time PCR (TqPCR) master mixes. A = SsoAdvanced™ Universal Probes Supermix (Bio-Rad) 60 °C; B = SsoAdvanced™ Universal Probes Supermix (Bio-Rad) 63 °C; C = QuantiNova pathogen+IC kit (QIAGEN S.r.l); D = Fast Universal PCR Master Mix (Applied Biosystems); E = Brilliant multiplex QPCR Master Mix (Agilent) *Xf = Xylella fastidiosa*; *Xfm = Xylella fastidiosa* subspecies *multiplex*; *Xff = Xylella fastidiosa* subspecies *fastidiosa*; *Xfp = Xylella fastidiosa* subspecies *pauca*.

| TqPCR Mastermix | Performance Criteria (%) | *Xf* | *Xff* | *Xfm* | *Xfp* |
|---|---|---|---|---|---|
| A | DSE | 97.91 | 100 | 100 | 91.66 |
|  | DSP | 100 | 100 | 100 | 94.58 |
|  | ACC | 98.61 | 100 | 100 | 91.66 |
| B | DSE | 95 | 100 | 100 | 87.5 |
|  | DSP | 100 | 100 | 100 | 100 |
|  | ACC | 96.6 | 100 | 100 | 98 |
| C | DSE | 87.5 | 87.5 | 87.5 | 100 |
|  | DSP | 93.7 | 100 | 100 | 100 |
|  | ACC | 89.6 | 97.9 | 95.8 | 100 |
| D | DSE | 100 | 100 | 100 | 100 |
|  | DSP | 100 | 100 | 100 | 100 |
|  | ACC | 100 | 100 | 100 | 100 |
| E | DSE | 100 | 100 | 100 | 100 |
|  | DSP | 100 | 100 | 100 | 100 |
|  | ACC | 100 | 100 | 100 | 100 |

**Table 6.** Non-concordant values obtained with the TqPCR master mixes used for the test performance study by the participating official laboratories. FP = number of false positives; FN = number of false negatives; A = SsoAdvanced™ Universal Probes Supermix (Bio-Rad) (60 °C); B = SsoAdvanced™ Universal Probes Supermix (Bio-Rad) (63 °C); C = QuantiNova pathogen+IC kit (QIAGEN S.r.l); D = Fast Universal PCR Master Mix (Applied Biosystems); E = Brilliant multiplex QPCR Master Mix (Agilent); *Xf = Xylella fastidiosa*; *Xfm = Xylella fastidiosa* subspecies *multiplex*; *Xff = Xylella fastidiosa* subspecies *fastidiosa*; *Xfp = Xylella fastidiosa* subspecies *pauca*.

| TqPCR Master Mix | *Xf* | | *Xff* | | *Xfm* | | *Xfp* | |
|---|---|---|---|---|---|---|---|---|
|  | FP | FN | FP | FN | FP | FN | FP | FN |
| A | 0 | 1 | 0 | 0 | 0 | 0 | 1 | 1 |
| B | 0 | 2 | 0 | 0 | 0 | 0 | 0 | 1 |
| C | 1 | 4 | 0 | 1 | 0 | 2 | 0 | 0 |
| **D** | 0 | 0 | 0 | 0 | 0 | 0 | 0 | 0 |
| **E** | 0 | 0 | 0 | 0 | 0 | 0 | 0 | 0 |

Percentage values of performance criteria below 100% were obtained by seven OLs (Figure S1). As shown in Table 6, two OLs of Group A gave inconsistent results: one OL obtained two false negatives in identifying *Xf* and *Xf* subspecies *pauca* (S 6), and the other OL gave a false positive for the subspecies *pauca* (S 12). In group B, one OL gave three false negative values for *Xf* (S 5 and S 11) and *Xf* subspecies *pauca* (S 11). In group C, only one OL showed a non-concordant result, obtaining one false positive (S 2) and four false negatives (S 4, S 5, S 6, and S 9) for *Xf*, one false negative (S 5) for *Xf* subspecies *fastidiosa*, and two false negatives for *Xf* subspecies *multiplex* (S 4 and S 9).

The percentages of reproducibility (CO) are reported in Table 7, and the values are between 87.5 and 100% (Table 7).

**Table 7.** Reproducibility values of the different TqPCR master mixes used for the test performance study for samples S 1, S 2, S 4, S, S 6, S 7, and S 8. (A = SsoAdvanced™ Universal Probes Supermix (Bio-Rad) 60 °C; B = SsoAdvanced™ Universal Probes Supermix (Bio-Rad) (63 °C); C = Quanti-Nova pathogen+IC kit (QIAGEN S.r.l); D = Fast Universal PCR Master Mix (Applied Biosystems); E = Brilliant multiplex QPCR Master Mix (Agilent). Xf = *Xylella fastidiosa*; Xfm = *Xylella fastidiosa* subspecies *multiplex*; Xff = *Xylella fastidiosa* subspecies *fastidiosa*; Xfp = *Xylella fastidiosa* subspecies *pauca*.

| TqPCR Master Mix | Xf | Xff | Xfm | Xfp |
|---|---|---|---|---|
| A | 95.83 | 100 | 100 | 95.83 |
| B | 95 | 100 | 100 | 100 |
| C | 95 | 93.75 | 87.5 | 100 |
| D | 100 | 100 | 100 | 100 |
| E | 100 | 100 | 100 | 100 |

Non-concordant results for reproducibility determination were evaluated. In particular, two OLs obtained false negative values, the details of which are shown in Table 8. In group A, one OL presented two false negative results for *Xf* and *Xf* subspecies *pauca* (S 6); in group B, one false negative result was recorded for *Xf* (S 5); and in group C, one false positive result (S 2) and three false negatives (S 4, S 5, and S 6) were reported for *Xf*, one false negative (S 5) was reported for *Xf* subspecies *fastidiosa*, and one false negative (S 4) was reported for *Xf* subspecies *multiplex*. The overall results showed that a lower concordance was obtained by group C, which used the QuantiNova pathogen+IC kit (Qiagen S.r.l.). Moreover, a high number of non-concordant results (mostly false negatives) occurred with primers/probes for *Xf* detection (6/9) with respect to those used for subspecies identification (3/9) (Table 8).

**Table 8.** Reproducibility: non-concordant values obtained with the TqPCR master mixes used by different participating laboratories. FP = number of false positives; FN = number of false negatives; A = SsoAdvanced™ Universal Probes Supermix (Bio-Rad) (60 °C); B = SsoAdvanced™ Universal Probes Supermix (Bio-Rad) (63 °C); C = QuantiNova pathogen+IC kit (QIAGEN S.r.l); D = Fast Universal PCR Master Mix (Applied Biosystems); E = Brilliant multiplex QPCR Master Mix (Agilent). Xf = *Xylella fastidiosa*; Xfm = *Xylella fastidiosa* subspecies *multiplex*; Xff = *Xylella fastidiosa* subspecies *fastidiosa*; Xfp = *Xylella fastidiosa* subspecies *pauca*.

| TqPCR Master Mix | Xf | | Xff | | Xfm | | Xfp | |
|---|---|---|---|---|---|---|---|---|
| | FP | FN | FP | FN | FP | FN | FP | FN |
| A | - | 1 | - | - | - | - | - | 1 |
| B | - | 1 | - | - | - | - | - | - |
| C | 1 | 3 | - | 1 | - | 1 | - | - |
| D | - | - | - | - | - | - | - | - |
| E | - | - | - | - | - | - | - | - |

### 3.1.4. Comparison of Real-Time PCR and Tetraplex Real-Time PCR

A comparison (Table S4) of the Cq values obtained via TqPCR [16] with those obtained using the reference test developed by Harper et al. [18] showed that the Cq values were comparable (Figure 1).

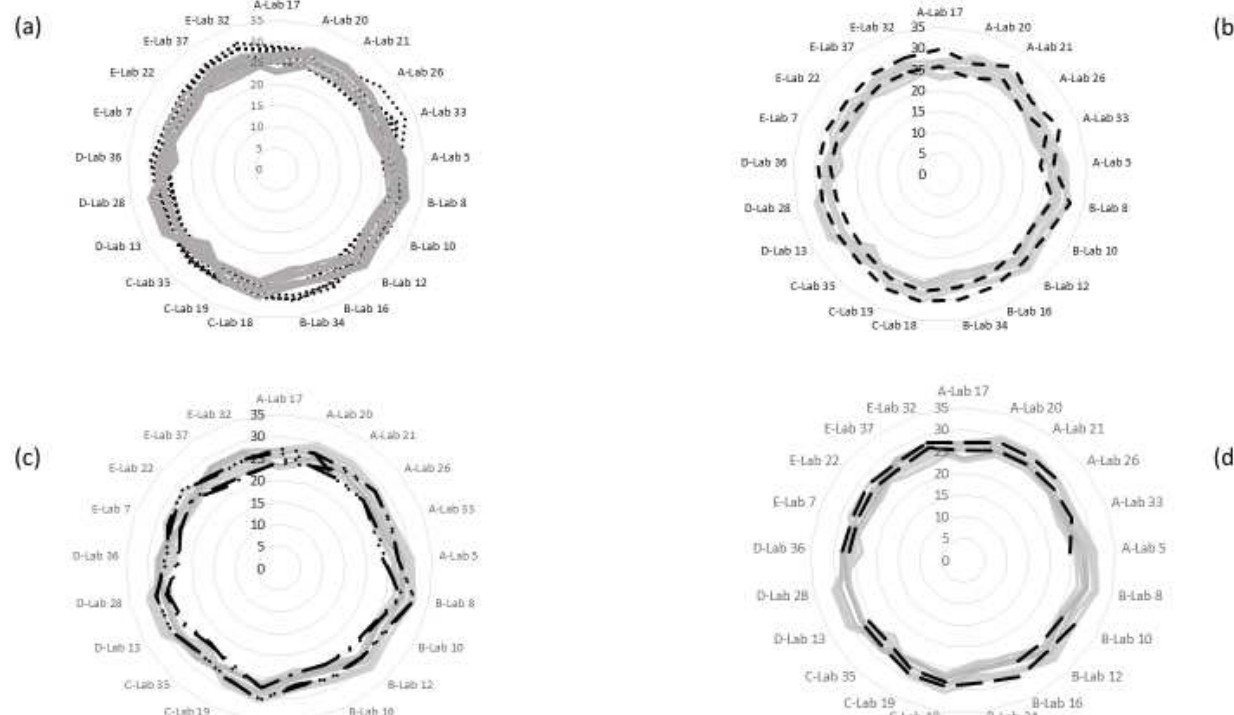

**Figure 1.** Summary of the quantification cycle (Cq) values obtained by participating official laboratories within the test performance study. The Cq values obtained using the Harper et al. [18] test, represented by the solid gray line, are compared with the Cq values obtained using TqPCR (Dupas et al. [16]), indicated with a dotted line; the line is missing when the laboratory did not perform the analysis with the respective fluorophore. (**a**) FAM fluorophore for *Xylella fastidiosa*; (**b**) HEX fluorophore for *Xylella fastidiosa* subspecies *fastidiosa*; (**c**) ROX fluorophore for *Xylella fastidiosa* subspecies *multiplex*; (**d**) Cy5 fluorophore for *Xylella fastidiosa* subspecies *pauca*.

### 3.2. Intralaboratory Study (ITS)

The results of the real-time PCR conducted on the spiked samples are reported in Table 9, in which the ASE values are indicated. ASE was estimated at $10^2$ CFU mL$^{-1}$ for Harper et al.'s [18] and Hodgetts et al.'s [20] assays for all the matrices and *Xf* subspecies, except for the combination *O. europaea/Xfp*, for which the value decreased to $10^3$ CFU mL$^{-1}$. For TqPCR, ASE showed a range of $10^2$–$10^4$ CFU mL$^{-1}$ depending on the combination of plant matrix/*Xf* subspecies/master mix used. With the master mix of Bio-Rad, the sensitivity values decreased to $10^4$ CFU mL$^{-1}$, except for the detection of *Xfp* in *O. europaea*, for which value of $10^3$ CFU mL$^{-1}$ was reported. The master mix of Agilent showed the best performance criteria in the TPS activity, reporting an ASE value of $10^2$ CFU mL$^{-1}$ for the detection of *Xf* subspecies *fastidiosa* in *V. vinifera* (for both probes of *Xf* and *Xf* subspecies *fastidiosa*), $10^3$ CFU mL$^{-1}$ for *Xf* subspecies *multiplex* in *P. dulcis* (for both probes *Xf* and *Xf* subspecies *multiplex*), and $10^4$ CFU mL$^{-1}$ and $10^2$ CFU mL$^{-1}$ for the detection of *Xf* and *Xfp* in *O. europaea*, respectively. In general, it should be noted that the limit of detection obtained by using the Biorad master mix was lower than that obtained using the Agilent master mix (Table 9).

**Table 9.** Results obtained using real-time PCR (Harper et al. [18], Hodgetts et al. [20], and Dupas et al. [16]) with SsoAdvanced™ Universal Probes Supermix (Bio-Rad) and Brilliant Multiplex qPCR Master Mix (Agilent) in matrices of *Olea europaea* (*Oe*), *Vitis vinifera* (*Vv*), and *Prunus dulcis* (*Pd*) spiked with *Xfp*, *Xff*, and *Xfm*, respectively. For each test, the average of Cq values (Cq), the number of positive replicates of the total assessed samples (*N*), and standard deviation (SD) are reported. The values in bold indicate the analytical sensitivity (ASE) of each method for the different tested matrix/*Xf* subspecies combinations. *Xf* = *Xylella fastidiosa*; *Xfm* = *Xylella fastidiosa* subspecies *multiplex*; *Xff* = *Xylella fastidiosa* subspecies *fastidiosa*; *Xfp* = *Xylella fastidiosa* subspecies *pauca*.

| Real-Time Test | Plant Matrix/*Xf* Subspecies | Sub Species | $10^6$ N | $10^6$ Cq | $10^6$ DS | $10^5$ N | $10^5$ Cq | $10^5$ DS | $10^4$ N | $10^4$ Cq | $10^4$ DS | $10^3$ N | $10^3$ Cq | $10^3$ DS | $10^2$ N | $10^2$ Cq | $10^2$ DS | $10^1$ N | $10^1$ Cq | $10^1$ DS |
|---|---|---|---|---|---|---|---|---|---|---|---|---|---|---|---|---|---|---|---|---|
| Harper et al. [18] | O/*Xfp* | | 9/9 | 24.97 | 0.44 | 9/9 | 29.02 | 0.54 | 9/9 | 32.08 | 0.16 | **9/9** | 35.48 | 0.74 | 8/9 | 37.86 | 0.82 | 4/9 | 38.95 | 0.44 |
| Harper et al. [18] | *Vv*/*Xff* | *Xf* | 9/9 | 23.58 | 0.46 | 9/9 | 26.28 | 0.45 | 9/9 | 28.47 | 1.22 | 9/9 | 31.76 | 1.40 | **9/9** | 34.82 | 1.38 | 7/9 | 36.51 | 1.32 |
| Harper et al. [18] | *Pd*/*Xfm* | | 9/9 | 25.75 | 0.48 | 9/9 | 29.74 | 0.20 | 9/9 | 32.90 | 1.19 | 9/9 | 35.60 | 0.80 | **9/9** | 36.52 | 0.98 | 3/9 | 36.03 | 1.57 |
| Hodgetts et al. [20] | O/*Xfp* | *Xf* | 9/9 | 24.13 | 0.46 | 9/9 | 27.46 | 0.46 | 9/9 | 30.22 | 0.43 | **9/9** | 33.28 | 0.39 | 8/9 | 35.94 | 0.78 | 6/9 | 37.01 | 0.44 |
| Hodgetts et al. [20] | *Vv*/*Xff* | *Xff* | 9/9 | 23.12 | 0.47 | 9/9 | 26.06 | 0.41 | 9/9 | 29.19 | 0.32 | 9/9 | 32.40 | 0.47 | **9/9** | 35.06 | 0.55 | 6/9 | 36.91 | 0.62 |
| Hodgetts et al. [20] | *Pd*/*Xfm* | *Xfm* | 9/9 | 24.09 | 0.38 | 9/9 | 28.34 | 0.19 | 9/9 | 31.28 | 0.78 | 9/9 | 34.04 | 0.46 | **9/9** | 36.10 | 1.60 | 8/9 | 35.24 | 1.98 |
| Dupas et al. [16] — Bio-Rad | O/*Xfp* | *Xf* | 9/9 | 28.91 | 0.51 | 9/9 | 31.71 | 0.60 | **9/9** | 36.06 | 1.05 | 8/9 | NA | NA | 0/9 | NA | NA | 0/9 | NA | NA |
| Dupas et al. [16] — Bio-Rad | O/*Xfp* | *Xfp* | 9/9 | 25.57 | 0.54 | 9/9 | 28.64 | 0.43 | 9/9 | 31.70 | 0.55 | **9/9** | 36.23 | 0.86 | 3/9 | NA | NA | 0/9 | NA | NA |
| Dupas et al. [16] — Bio-Rad | *Vv*/*Xff* | *Xf* | 9/9 | 28.06 | 0.73 | 9/9 | 31.96 | 0.82 | **9/9** | 38.39 | 0.78 | 0/9 | NA | NA | 0/9 | NA | NA | 0/9 | NA | NA |
| Dupas et al. [16] — Bio-Rad | *Vv*/*Xff* | *Xff* | 9/9 | 25.04 | 0.49 | 9/9 | 28.68 | 0.20 | **9/9** | 33.49 | 1.90 | 6/9 | NA | NA | 0/9 | NA | NA | 0/9 | NA | NA |
| Dupas et al. [16] — Bio-Rad | *Pd*/*Xfm* | *Xf* | 9/9 | 29.65 | 0.72 | 9/9 | 36.04 | 1.46 | **9/9** | NA | NA | 0/9 | NA | NA | 0/9 | NA | NA | 0/9 | NA | NA |
| Dupas et al. [16] — Bio-Rad | *Pd*/*Xfm* | *Xfm* | 9/9 | 26.04 | 0.47 | 9/9 | 30.05 | 0.22 | **9/9** | 34.29 | 1.67 | 6/9 | NA | NA | 0/9 | NA | NA | 0/9 | NA | NA |
| Dupas et al. [16] — Agilent | O/*Xfp* | *Xf* | 9/9 | 26.30 | 0.45 | 9/9 | 29.48 | 0.50 | **9/9** | 31.91 | 0.32 | 5/9 | 35.60 | 0.66 | 0/9 | NA | NA | 0/9 | NA | NA |
| Dupas et al. [16] — Agilent | O/*Xfp* | *Xfp* | 9/9 | 23.99 | 0.47 | 9/9 | 27.28 | 0.46 | 9/9 | 29.78 | 0.46 | **9/9** | 33.07 | 1.05 | 1/9 | 38.94 | NA | 0/9 | NA | NA |
| Dupas et al. [16] — Agilent | *Vv*/*Xff* | *Xf* | 9/9 | 25.46 | 0.45 | 9/9 | 28.15 | 0.33 | 9/9 | 31.20 | 0.29 | **9/9** | 34.96 | 1.84 | 0/9 | NA | NA | 0/9 | NA | NA |
| Dupas et al. [16] — Agilent | *Vv*/*Xff* | *Xff* | 9/9 | 23.74 | 0.42 | 9/9 | 26.45 | 0.36 | 9/9 | 29.59 | 0.41 | 9/9 | 32.75 | 0.29 | **9/9** | 35.77 | NA | 2/9 | NA | NA |
| Dupas et al. [16] — Agilent | *Pd*/*Xfm* | *Xf* | 9/9 | 27.32 | 0.33 | 9/9 | 31.23 | 0.14 | 9/9 | 34.22 | 1.10 | **9/9** | 35.23 | NA | 1/9 | NA | NA | 0/9 | NA | NA |
| Dupas et al. [16] — Agilent | *Pd*/*Xfm* | *Xfm* | 9/9 | 24.26 | 0.37 | 9/9 | 28.16 | 0.13 | 9/9 | 31.12 | 0.87 | **9/9** | 34.51 | 1.08 | 6/9 | 36.02 | NA | 0/9 | 35.64 | 1.68 |

The subspecies *pauca* and *multiplex* were identified in the naturally infected samples collected in the Apulia and Latium regions via TqPCR [16] performed with Brilliant Multiplex qPCR Master Mix (Agilent). These samples analyzed with the method developed by Harper et al. [18] showed Cq values between 20 and 29, which corresponded to an approximate bacterial concentration of $10^6$–$10^7$ CFU mL$^{-1}$. The allelic profile obtained via MLST analysis confirmed the result of the TqPCR, showing that the samples collected in Apulia belonged to *Xf* subspecies pauca ST53, whereas the samples collected in Lazio belonged to *Xf* subspecies multiplex ST87.

## 4. Discussion

The identification of *Xf* at the infraspecific level is decisive for the management of new outbreaks and essential for epidemiological study and surveillance. In fact, the host range of this pathogen and the list of plant species to be eradicated to limit the spread of the pathogen depend on the *Xf* subspecies. All plant hosts of the identified *Xf* subspecies within a newly infected area should be removed from the demarcated area (consisting of an infected zone and a buffer zone) [10]. Subspecies and ST identification is also crucial for the study of the origin of this pathogen and its population dynamics. Several tests for identifying *Xf* subspecies have been reported in the EPPO Protocol PM 7/24 (5) [11]; however in the

Annex IV of the EU regulation 2020/1201 [10] there are currently no available methods for identifying all subspecies, except for the MLST based on Yuan et al. [14]. This scheme is based on amplification via conventional PCR and the sequencing of seven loci (*cysG*, *gltT*, *holC*, *leuA*, 242 *malF*, *nuoL*, and *petC*). For each locus, the different sequence variants are considered as distinct alleles. The combination of allele numbers defines the ST. The *MLST-Xf* data are stored in a public database (https://pubmlst.org/xfastidiosa/, accessed on 17 January 2023) that can be used to automatically identify and assign new allele variants.

In case of new findings, outbreaks, and hosts, the MLST schemes [14] based on the amplification and sequencing of two housekeeping genes for subspecies identification and of seven housekeeping genes for ST identification are required; however, these procedures are labor-intensive, time consuming, and expensive, and while successful when using DNA from pure bacterial culture, they are less reliable with DNA from plant extracts with low yields of DNA or in the case of mixed infections [15]. Recently, a nested-MLST assay was developed [25] to improve the sensitivity of *Xf* identification applied directly to plant-extracted DNA, showing a limit of detection very similar to that reported by Harper et al. [18]. It is worth noting that the main limitation of the nested PCR technique is the high risk of cross-contamination between samples due to the addition of the extra step of amplification. Other than TqPCR [16], another test able to detect all subspecies is the real-time assay developed by Hodgetts et al. [20], which requires single reactions for each subspecies' identification. With respect to these latter tests, the TqPCR method developed by Dupas et al. [16] allows, in one step, *Xf* detection and subspecies identification directly from plant samples and needs less template DNA and fewer consumables and reagents.

The results of the TPS carried out with the participation of 18 OLs of the Italian NPPO showed acceptable performance values for the TqPCR with Set N° 2 of primers and probes for all the critical reagents tested. The best results, with 100% performance criteria values, were obtained using Fast Universal PCR Master Mix (Applied Biosystems) and Brilliant multiplex QPCR Master Mix (Agilent) (group D and E, respectively). Small variations from the expected results were obtained using the master mix SsoAdvanced™ Universal Probes Supermix (Bio-Rad) employed at a 60 °C annealing temperature and in accordance with the procedure described in EPPO PM7/24 (5). In particular, with this master mix, at both 60 °C and 63 °C annealing temperatures (Group A and B), some false positive and false negative results were obtained (Tables 7 and 9). The lowest ACC (89.6%) was observed when the master mix QuantiNova pathogen+IC kit (QIAGEN S.r.l) (Group C) was used.

Evaluating the results of those OLs that performed both the reference test developed by Harper et al. [18] and TqPCR [16] with the same instrument, conditions, and operator, the comparison of the Cq values showed minor variations (Figure 1). It is well known that the qPCR method of Harper et al. [18] is one of the most reliable and sensitive tests for the detection of *Xf* [19,26] in a wide range of host plants. Our results confirm that TqPCR is reliable for samples at a medium-high level of infection, offering the advantage of simultaneously identifying the subspecies. It should be noted that in the tetraplex reaction, false results are sometimes associated with primers/probes for *Xf* species identification, although subspecies-specific primers/probes are reliable, producing a low number of false results (Table 8). Furthermore, the selection of the most suitable enzyme for the amplification reactions completely reduces the risk of obtaining false results.

Previous studies showed that the average value of the limit of detection of the MLST assay was about $10^5$ CFU mL$^{-1}$ [14,17], while TqPCR was reported to exhibit a detection limit next to the reference protocol of Harper et al. [18], that is, $10^2$–$10^4$ CFU mL$^{-1}$ [16,18]; in our conditions, it allowed for *Xf* detection in all spiked matrices at up to $10^3$ CFU mL$^{-1}$. If the qPCR method of Harper et al. [18] remains the gold standard for *Xf* detection, in case of positive detection, TqPCR [16] could be useful for confirming positivity and identifying the subspecies.

It is worth noting that, in the case of new outbreaks or new presumptive plant hosts, NPPOs require National Reference Laboratories to perform a confirmatory analysis to confirm the detection of *Xf* and to identify the subspecies. A possible workflow should be

the use of a detection test, i.e., Harper et al.'s test [18] and the application of TqPCR [16], for confirmation and subspecies identification, with the latter serving as an alternative to the MLST assay [14], which is very costly and time consuming.

The ITS for the ASE evaluation highlighted a sensitivity of $10^2$–$10^3$ CFU mL$^{-1}$ for the methods developed by Harper et al. [18] and Hodgetts et al. [20] and $10^2$–$10^4$ CFU mL$^{-1}$ for TqPCR, [16] depending on the enzyme used in the reaction and the plant matrices considered. The Agilent master mix, under our conditions, showed better results than the Biorad master mix, showing ASE values comparable to the other two methods [18,20] for the analysis of *V. vinifera* and *P. dulcis*. These results show that both the Hodgetts [20] and Dupas [16] methods are valid methods for confirming the results obtained with Harper et al.'s method [18] and can help in the identification of the subspecies without the bacterial isolation step.

The analyses of the naturally infected samples of *P. dulcis* and *S. junceum*, collected in Latium and *O. europea* samples collected in Apulia, showed that the results of the MLST [14] and the TqPCR [16] correctly assigned the subspecies *pauca* (*O. europaea*—Apulia region) and *multiplex* (*P. dulcis* and *S. junceum*—Latium) with both techniques, thus confirming the reliability of subspecies assignment, with a considerable advantage in terms of the time required for the TqPCR [16]. In fact, the TqPCR identifies the subspecies in a single step, whereas the MLST procedure requires the following steps: conventional PCR, electrophoresis, an amplicon purification step, sequencing, the determination of the consensus sequence for each target gene, and in silico analysis using the pubMLST database. These results confirm the effectiveness of the TqPCR method when applied to naturally infected samples, as previously evidenced by Dupas et al., [16] during the development phase of this method.

Based on the results obtained in this validation study, it can be speculated that the inclusion of TqPCR [16] among the tests to be used by laboratories performing an official diagnosis of *Xf* could provide great support for phytosanitary activities. Indeed, the EPPO protocol recently included this method in the fifth revision of the PM7/24 (5) [11]. Since the tests reported in Annex 4 of the EU regulation 2020/1201 [10] are mandatory for official analyses, it is advisable that these tests be considered for inclusion. The TPS results showed that with the used DNA extraction method (DNAesy Mericon Food Kit (Qiagen), with different master mixes, instruments, laboratory conditions, and different manual skills, TqPCR [16] proves to be a robust method that confirms subspecies identification in different matrices in samples with a medium-high level of infection. Moreover, compared with the MLST technique [14] for subspecies determination, TqPCR [16] saves time and money. Our results on spiked samples showed that TqPCR [16], with Brilliant Multiplex qPCR Master Mix (Agilent), was suitable for the diagnosis and identification of *Xf* subspecies with a sensitivity similar to that of the Harper et al. [18] and Hodgetts et al. [20] tests. Our preliminary data on the ASE obtained using the test developed by Hodgett et al. [20] suggest that this test could also to be very useful for subspecies identification, but it requires the preparation of different master mixes for each subspecies identification along with a greater use of consumables and staff to perform the real-time PCR.

Concerning the ST identification of naturally infected samples of *P. dulcis* and *S. junceum*, it was shown that they belong to ST87, the same clonal complex of *Xf* subspecies *multiplex* recovered in Tuscany [27–29]; indeed, the allelic sequences in each gene locus showed 100% identity with allelic variants described in the database and associated with ST 87 and ST53. These results suggest that this strain might have been imported from the adjacent area of Argentario (Tuscany region), where it is widespread [29], indicating that this clonal population seems to be expanding or to be established in central Italy; further study may help to extend and implement the genomic data available for isolates harboring ST 87. It should be noted that the subspecies *multiplex* seems to be permanently established in the Mediterranean areas in association with endemic species of the typical flora, constituting a situation like that in Spain and France, where infected natural settings (shrubs, weeds, and ornamentals) are widely represented [15,30]. The allelic profiles

obtained for the olive samples collected in Apulia corresponded to ST53, which belongs to the clonal complex of *Xf* subspecies *pauca* [31], the only strain widely spread in Apulia.

**Supplementary Materials:** The following are available online at https://www.mdpi.com/article/10.3390/horticulturae9091053/s1. Figure S1: Qualitative results obtained by labs participating in the test performance study. In gray are indicated the results not in accordance with the expected phytosanitary status. Legend: pos = positive; neg = negative; NA = non evaluable; POLs = participating official laboratories; Xfm *Xylella fastidiosa* subspecies *multiplex*; Xff = *Xylella fastidiosa* subspecies *fastidiosa*; Xfp = *Xylella fastidiosa* subspecies *pauca*; Table S1: Tetraplex qPCR of Dupas et al., [16]: reagents and master-mix used by the labs participating in the test performance study divided in five groups: A = SsoAdvanced™ Universal Probes Supermix (Bio-Rad) 60 °C; B = SsoAdvanced™ Universal Probes Supermix (Bio-Rad) 63 °C; C = QuantiNova pathogen+IC kit (QIAGEN S.r.l); D = Fast Universal PCR Master Mix (Applied Biosystems); E = Brilliant multiplex QPCR Master Mix (Agilent). Primers and probes legenda: 1-XF to detect *Xylella fastidiosa* 2-XFF to detect *Xylella fastidiosa* subspecies *fastidiosa*; 3-XFM to detect *Xylella fastidiosa* subspecies *multiplex*; 4- XFP to detect *Xylella fastidiosa* subspecies *pauca*; Table S2: Evaluation of homogeneity values of test performance study samples using Real-Time PCR (Harper et al., [18]). The phytosanitary status of each sample, the number of positive samples over the total number of samples tested (pos/tot), the maximum and minimum Cq values (cycle threshold) obtained for each sample, the averages of the Cq values and the respective standard deviation (SD) are reported. NA = not amplified. Three panels were tested for each sample in technical duplicate. Xfm = *Xylella fastidiosa* subspecies *multiplex*; Xff = *Xylella fastidiosa* subspecies *fastidiosa*; Xfp = *Xylella fastidiosa* subspecies *pauca*.; Table S3: Evaluation of stability of test performance study samples using Real-time PCR (Harper et al., [16]) after 7 days of storage at <−15 °C, 2–8 °C and 25 °C (mid-term stability) and at the end of the study (long-term stability) after 4 weeks. Three panels were tested for each sample in technical duplicate. The phytosanitary status of each sample, the number of positive samples over the total number of samples tested (pos/tot), the maximum and minimum values of the Cq (cycle threshold) obtained for each sample, the averages of the Cq values and the respective standard deviation (SD) are reported. NA = not amplified; Xfm = *Xylella fastidiosa* subspecies *multiplex*; Xff = *Xylella fastidiosa* subspecies *fastidiosa*; Xfp = *Xylella fastidiosa* subspecies *pauca*; Ac = artficially contamined; Table S4: Results obtained in Real-Time PCR according to Harper et al. [18] and in tetraplex Real-Time PCR (Dupas et al. [16]) concerning *Xylella fastidiosa* identification. The phytosanitary status of each sample, the averages of the Cq values and the respective standard deviation value is reported. Legend: Cq = threshold cycle; SD = standard deviation; NA not amplified Xfm = *Xylella fastidiosa* subspecies *multiplex*; Xff = *Xylella fastidiosa* subspecies *fastidiosa*; Xfp = *Xylella fastidiosa* subspecies *pauca*.

**Author Contributions:** Conceptualization, N.P., V.S. and S.L. (Stefania Loreti); methodology, E.B., E.C., V.C., R.F., A.L., S.L. (Simone Lucchesi), G.T., S.C., F.D.A., S.F., F.G. (Francesca Giacobbi), F.G. (Francesca Gaffuri), V.G., L.L., G.L., G.M., S.P., M.L.R., D.R., C.R., M.R.S., S.T. and M.T.; validation, N.P. and S.L. (Stefania Loreti); data curation, N.P., A.L. and G.T.; writing—original draft preparation, N.P.; writing—review, editing and supervision N.P., V.S. and S.L. (Stefania Loreti); funding acquisition, S.L. (Stefania Loreti). All authors have read and agreed to the published version of the manuscript.

**Funding:** This research was funded by MIPAAF, Proteggo 1.4 "DISR-05-0001837-04/01/2022 Ministero delle politiche agricole alimentari e forestali–Consiglio per la ricerca in agricoltura e l'analisi dell'economia Agraria".

**Data Availability Statement:** All data are presented within the article.

**Conflicts of Interest:** The authors declare no conflict of interest.

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
