# Peer review of "An Inter-Laboratory Comparative Study on the Influence of Reagents to Perform the Identification of the Xylella fastidiosa Subspecies Using Tetraplex Real Time PCR"

_horticulturae, doi:10.3390/horticulturae9091053_

Round 1
Reviewer 1 Report
The manuscript, entitled: "The Inter-laboratory comparison study on the influence of reagents to perform the Xylella fastidiosa subspecies identification through the Tetraplex Real Time PCR" is a summary of a validation study compiled with due care and precisely executed and evaluated.
Xylella fastidiosa (Xf) is a very important gram-negative bacterium that infects the xylem parts of more than 600 host plants and is the cause of very significant plant diseases [e.g. g. Pierce's disease in grapes, variegated chlorosis of citrus, olive rapid decline syndrome (OQDS), phony peach disease]. It is spread by vector insects that feed on xylem sap. Xf has been on the EPPO A2 pathogen list since 2017.
This pathogen, which is already widespread in the Mediterranean region, threatens other European areas to a significant extent, and the predictions of warming models support this.
Knowledge of subspecies and sequence types (ST) is essential for effective protection and eradication. To determine these, the EPPO also prepared methodological recommendations.
This comprehensive, validation test series verifies the test results of 17 reference laboratories in Italy and the adequacy of the previously used (MLST) and the proposed new diagnostic method (TqPCR).
Preparation of tests and samples is thorough and precise. Quality control of samples is adequate. The statistical processing and interpretation of the obtained results is excellent.
In order to improve the quality of the manuscript, I suggest correcting some formal errors:
1.- In the case of group code "C", "D" and "E" in table no. 3 (on page 7 of the manuscript), the time and temperature data have shifted compared to the cycles data.
2. In the header of table no. 9 (on page 15 of the document), I recommend the standard notation DS for the deviation standard.
After implementing the changes suggested above, I recommend publishing the scientific article.
Reviewer 2 Report
The study was conducted to assess the impact of different reagents on the performance of the tetraplex Real-Time PCR for identifying Xylella fastidiosa subspecies. The study addresses an important aspect of molecular diagnostics in plant health, focusing on the influence of reagents on the accuracy and reliability of X. fastidiosa subspecies identification. The abstract establishes the context by mentioning the significance of the TqPCR method and its potential contribution to official monitoring activities.
It might be beneficial to include more details on the selection criteria for these laboratories, as well as the rationale behind the chosen mastermixes.
A brief explanation of the MLST method used for comparison could enhance understanding.
While the abstract mentions that the reagents with the best results from the test performance study (TPS) and intralaboratory study (ITS) were employed, the specific findings and performance metrics of these reagents are not provided. Including some indicative results, such as sensitivity and specificity values, could offer readers a glimpse into the practical outcomes of the study.
To enhance the practical implications, it would be valuable to briefly mention the alignment of results between the TqPCR method and the official MLST method. This would highlight the potential utility of the TqPCR approach in real-world diagnostics.
PD2 medium, add its composition
Some sentences could be rephrased for greater clarity. For instance, the sentence "The reagents that showed the best results in the TPS and ITS are applied to identify the subspecies on naturally infected samples collected from Italy in comparison with the official MLST method" could be reworded for smoother comprehension.
It might be useful to briefly discuss limitations or potential future directions to provide a well-rounded perspective.
Conclusion is missing
Minor editing is required
Reviewer 3 Report
The manuscript: “The Inter-laboratory comparison study on the influence of reagents to perform the Xylella fastidiosa subspecies identification through the Tetraplex Real Time PCR “(Manuscript ID: horticulturae-2561054). The tetraplex Real-Time PCR was used for detecting Xylella fastidiosa subspecies from Italy. The results of this study are important, but this manuscript must be removed. Then, I report some points as below.
1. The authors used only twelve blind samples, I thinking it too few.
2. Reference in Abstract should be deleted.
3. 1×104 should be 1×104 in line 104, authors must check it carefully in all text.
4. in lines 117-124, different formats of reference.
5. in Table S2, “μl” should be “μL”.
Reviewer 4 Report
The manuscript entitled " The Inter-laboratory comparison study on the influence of reagents to perform the Xylella fastidiosa subspecies identification through the Tetraplex Real Time PCR” reports the results of an inter-laboratory comparative study aiming at testing the impact of 5 conditions (Taq polymerases included in mastermix and cycle temperature) on the detection and identification of 3 subspecies of X. fastidiosa using one of the multiplex qPCR assay developped by Dupas et al. (2019). This independent study is of importance to demonstrate the potential of this novel multiplex qPCR and can form a reference for other NPPOs. Overall this paper fits in the scope of this journal. I have some comments that should be considered before publication of this MS.
-Main comments:
- The authors organized an inter-laboratory Test Performance Study for the Dupas’ test and compared the LODs of Dupas’ test to the Hodgetts’ test in a Intra-Laboratory study. Surprisingly in the discussion section they made recommendations at a similar level for both tests (L. 504-507). This would suggest that the test performance study, which requested the participation of 17 laboratories, has no value. The Authors should stick to the aim of their study (L. 112-124) and make recommendation in the Discussion section only for the Dupas’ test, considering that the Hodgetts test was used as a comparative test.
-The authors chose the Tetraplex set N°2 targeting XF–XFF–XFM–XFP. The choice of this set implies that they have chosen to target the X. fastidiosa subsp. fastidiosa (Xff in Dupas et al. paper) and not X. fastidiosa subsp. fastidiosa sensu largo (XFFSL in Dupas et al paper), hence the sandyi and morus lineages cannot be identified. This is a choice and I suggest that the Authors clarify it in their manuscript. For example a sentence can be added in L. 111 to precise this point.
-The authours used the present tense in most parts of the manuscript (Abstract, Materials and Methods, and Results sections), when it is generally recommended to stick to the past tense. The widespread use in this manuscript of a passive form may perhaps explain this confusion, but it is important to use the past tense in these sections. I therefore recommend that the authors correct the manuscript in this sense, and possibly change some of the cumbersome passive forms with more active ones.
-In Dupas et al., 2019, a sonication step was added to improve DNA extraction from plant sample. This was apparently not done in this study. The Authors shall mention this difference with regard to the original description of the method.
-In the abstract the final sentence that consists in a recommendation is a little be awkward. The Authors should write a clearer recommendation (by way of example, and sticking as closely as possible to the original sentence, I would suggest “The results obtained in this comparative study indicate that the TqPCR assay of Dupas et al. (2019) can figure among the tests to be used by laboratories performing official diagnosis of Xf to support the activities of official monitoring.”)
Minor comments:
Title: please change “comparison study” by “comparative study”
L. 47-48: please change “to the family of Lysobacteraceae (formerly Xanthmonadaceae)” by “to the of Lysobacteraceae (formerly Xanthmonadaceae) family”
L62: Xf has also been reported in Occitanie area in France, where eraidcation is ongoing (https://food.ec.europa.eu/plants/plant-health-and-biosecurity/legislation/control-measures/xylella-fastidiosa/latest-developments-xylella-fastidiosa-eu-territory_en)
L. 73: please change “previously free area” by “previously Xf-free area”
L. 74: please change “new plant host” by “new host plant”
L. 98 : please consider reformulating “for Xf is performed amplifying the plant DNA” by “can be performed for Xf by a direct amplifcation on DNA extracted from plant sample”.
L. 104-105: please pay attention to the formating of values: 1×103 - 1×104 ; 105
L. 106: please rephrase “and is like the sensitivity” by “and is similar to the sensitivity”
L 127-128: please consider changing “nucleic acid (DNA)” by “DNA samples”.
L131: “of Xf strains from the subspecies”
L 135: “of Xf strains from the subspecies”
L. 159: please change “areas Xf free” by “Xf-free area”
L. 190: please change “e” by “and”
Table 3: please correct the layout problems in the table (partial empty rows)
L. 219-220: please correct by “the percentage of true negative (TN), false positive (FP), false negative (FN) and true positive (TP) results”
L. 222: Which LOD was used? How was it defined?
L. 263: please change “were” by “was”
L; 272-273: Please consider rephrasing by “The results are in accordance with the phytosanitary status (healthy or contaminated) of the samples;”
L. 274: please change “variations” by “variation”
L. 275: please change “stables” by “stable”
Table 6: please fill in the empty cell (E - Xff-FP)
L. 401: “with respect to the values”
L. 407: please change “deviation standard” by “standard deviation”
Table 9 is not clear. Authors should add a colomn to clarify which test is used and a colomn for the mastermix. Please highlight the 9/9 result in the 104N - O/Xfp Xfp cell.
L. 443 please delete the “ that is not closed. Change “ It worth of noting» in « It is worth noting ».
L. 462-473: this part of the discussion was already made in Dupas et al., 2019 paper, and they suggest to change their Xf primers and probe by the Harper’s primers and probe and tested it. This should be discussed here.
L. 493: change “that both Hodgetts [20] that Dupas [16] are” by “that both Hodgetts [20] and Dupas [16] are”.
L. 504-507: as no interlaboratory comparative study was performed for the Hodgetts’ test, this recommendation should be tempered and should consider only Dupas’ test.
L. 525 and L. 533: it is clear to me why the Authors refer to “clonal complex” concerning ST87 or ST53. This should be clarified.
L. 526-528: the present results do not provide any evidence indicating that the ST87 strain present could have been imported from the Argentario area; This assertion would deserve much more analysis to determine the orientation of the migration and the adating of the outbreak. I suggest that the Authors rephrase this sentence.
Throughout the manuscript, I have suggested changes in wording or encourage to rephrase sentences. Alternatively, authors may prefer to use an independent English editing service.
Round 2
Reviewer 3 Report
Title : The Inter-laboratory comparison study on the influence of reagents to perform the Xylella fastidiosa subspecies identification through the Tetraplex Real Time PCR(Manuscript ID: horticulturae-2561054)
Authors must check some mistakes carefully in all text, regrettably, there are many unconventional writings in the revised manuscript.
Line 30: Dupas et al., (2019) should be deleted.
Line 105 and others: 10-100 pg.ml−1 should be 10−100 pg/mL. Here, pg/mL not pg.mL-1, i think the unit must keep correspondence with pg/µL in the Tables. Authors shouled be take notice of differencefor “−” and ”-” in all text and tables.
Line 486: 89,6% should be 89.6%.
Line 556: 27-28-29 should be 27−29.
In Table S1 Groupe D: 0.]?
References should be keep the same format.
Author Response
"Please see the attachment."
